# Network dynamics-based cancer panel stratification for systemic prediction of anticancer drug response

Minsoo Choi[1], Jue Shi[2], Yanting Zhu[2], Ruizhen Yang[2] & Kwang-Hyun Cho[1]

Cancer is a complex disease involving multiple genomic alterations that disrupt the dynamic response of signaling networks. The heterogeneous nature of cancer, which results in highly variable drug response, is a major obstacle to developing effective cancer therapy. Previous studies of cancer therapeutic response mostly focus on static analysis of genome-wide alterations, thus they are unable to unravel the dynamic, network-specific origin of variation. Here we present a network dynamics-based approach to integrate cancer genomics with dynamics of biological network for drug response prediction and design of drug combination. We select the p53 network as an example and analyze its cancer-specific state transition dynamics under distinct anticancer drug treatments by attractor landscape analysis. Our results not only enable stratification of cancer into distinct drug response groups, but also reveal network-specific drug targets that maximize p53 network-mediated cell death, providing a basis to design combinatorial therapeutic strategies for distinct cancer genomic subtypes.

[1] Department of Bio and Brain Engineering, Korea Advanced Institute of Science and Technology (KAIST), Daejeon 34141, Republic of Korea. [2] Center for Quantitative Systems Biology and Department of Physics, Hong Kong Baptist University, Hong Kong 999077, China. Minsoo Choi and Jue Shi contributed equally to this work. Correspondence and requests for materials should be addressed to K.-H.C. (email: ckh@kaist.ac.kr)

Cancer is a highly heterogeneous disease not only between disease types but also different patients with the same disease[1–3]. Cancer heterogeneity at the genomic level has been characterized by a number of comprehensive genome sequencing and molecular profiling analyses, and various computational methods were since developed to map the genomes of thousands of cancers to explain cancer complexity and identify opportunities for cancer prevention, early detection, and treatment[3,4]. For instance, large-scale genomic studies, such as The Cancer Genome Atlas (TCGA) and The Cancer Cell Line Encyclopedia (CCLE), have curated multi-level genomic information that can be further analyzed to understand variation in cancer genotypes and phenotypes[5–9]. In these studies, a large panel of cancer cell lines was profiled, using high-throughput measurements, such as genome sequencing, microarray, proteomics, and drug screening. In addition, the acquired large genomic data sets were used to establish a model to predict a relationship between drug sensitivity and genomic alterations of specific cancer cells as well as to identify response biomarkers to cancer therapeutics[6,10]. This approach is primarily based on analyzing genomic alterations at the molecular level and may help preclinical stratification of patients for more effective anticancer drug treatment. However, due to the complexity and often unknown effect of genomic alterations on actual dynamics and functions of specific cellular network/pathway, this individual molecule-based approach often falls short to provide comprehensive insight into the mechanistic origin of drug sensitivity and identify effective biomarker for drug response prediction.

Many research groups thus set out to develop alternative computational methods to analyze large genomic data sets based on cellular network topology, which consists of information of collective interactions between multiple components, such as genes and proteins, in an integrated manner. Compared to genomics analysis based on individual genomic alteration, the network topology-based approach is proven more effective to predict drug response (i.e., phenotype) from the genotypes[11], as well as classify and cluster cancer subtypes[12,13]. For example, method was developed to extract gene sub-networks from whole protein–protein interaction (PPI) network, based on which metastatic breast cancer was successfully classified[12]. Network-based stratification (NBS) was also successfully employed to classify cancers based on their mutation network profiles and demonstrated improved correlation between cancer subtypes and clinical outcomes[13]. However, effectiveness of these stratification methods is limited, as they often failed to predict clinical outcome of certain tumor subtypes that show clear clustering of genomic profiles[14]. This could partly be due to the fact that the performance of NBS analysis depends on the data type, which only took into account somatic mutation, but not methylation or copy number alteration (CNA), which likely also contributed to perturb the overall cellular responses. Moreover, as drug response is a highly dynamic process, classification of cancer subtypes based on only static network topology is evidently insufficient to identify biomarkers for predicting drug response. There is clearly a need to investigate dynamics of network and network perturbations at the system level to characterize and stratify cancer subtypes in terms of drug response.

Here, we present a network dynamics-based approach to systematically quantify how genomic alterations in cancer cells affect the function of biological networks and thus result in differential cellular phenotypes. Cancer cell can be viewed as a rewired network due to endogenous perturbations resulting from genomic alterations, which subsequently leads to modifications of signaling networks and their dynamic responses[11,15–18]. Such network rewiring is thought to be responsible for key oncogenic processes, such as uncontrolled proliferation and resistance to apoptosis

induced by both internal and external stimuli, e.g., drug[19]. Previous work by us and others showed functional states and dynamics of a cellular system of networks can be comprehensively studied by attractor landscape analysis[20–22]. Based on attractor landscape analysis of network dynamics, viable cellular phenotypes can be identified as steady states called attractor states. In this study, we extended the attractor landscape analysis of network to a large cancer cell panel by combining it with comprehensive genomic alteration profiles of these cancer cells to characterize cancer subtypes and developed a computational framework to evaluate drug efficacies and synergistic effects as a function of genotype.

We selected the p53 regulatory network for the attractor landscape analysis, given the importance of p53 network in regulating various aspects of cancer and anticancer drug response. Specifically, we first constructed differential p53 regulatory networks by mapping cancer genomics data from the CCLE database to a p53 network model and then analyzed their state transition dynamics under various perturbations that mimic the mechanism of drug action. Based on the network dynamics analysis, variable drug responses of the large cancer cell panel were categorized into distinct response subgroups. For each subgroup of cancer network response, we further investigated their specific p53 network dynamics and found that the differential p53 network dynamics determine the genotype-specific drug responses. Moreover, based on the network perturbation analysis, we also identified network-specific combinatorial targets that enhance particularly drug-induced cell death response and validated the computational prediction by experiments. Overall, our study established a novel computational framework to predict anticancer drug response based on cancer genotypes, which could be employed to design more effective, cancer-specific combinatorial therapy.

## Results

**Cancer cell stratification by attractor landscape analysis**. Our method of network dynamics-based stratification of human cancer cell panel integrates genomic alterations at multiple levels and is independent of tissue origin and cancer type. Specifically, cancer cell lines are described by differentially wired networks with distinct network topology resulted from their genomic alterations. The different cancer cell lines are subsequently clustered on the basis of their network dynamics in response to the same network perturbation, e.g., as a result of drug treatment. Network dynamics of the cell are analyzed by considering its attractor landscape, which consists of trajectories from all possible initial network states of the cell to its attractor states. We focus on the set of attractor states that different cancer cells eventually reach, which correspond to specific steady states of cellular phenotypes. In particular, for drug response, the viable steady-state attractor states include cell proliferation, cell cycle arrest, and cell death. Moreover, in the attractor landscape, area around each attractor state is the region of states with trajectories going to the attractor, which is called the "basin of attraction" or "basins", and can be used to measure the relative ratio of the respective cellular phenotypes.

Our systemic computational approach relies on three key steps: (1) selecting functional genomic alterations from a large number of molecular changes reported by the cancer genomics database; (2) constructing cancer cell-specific network models by mapping the functional genomic alterations of distinct cancer cell lines into the interaction network; and (3) stratifying cancer cells based on the network response profile to perturbations that can change the network dynamics. The functional genomic alterations that we selected include both somatic mutations and CNA that are associated with cancer development. For each cancer cell line, we

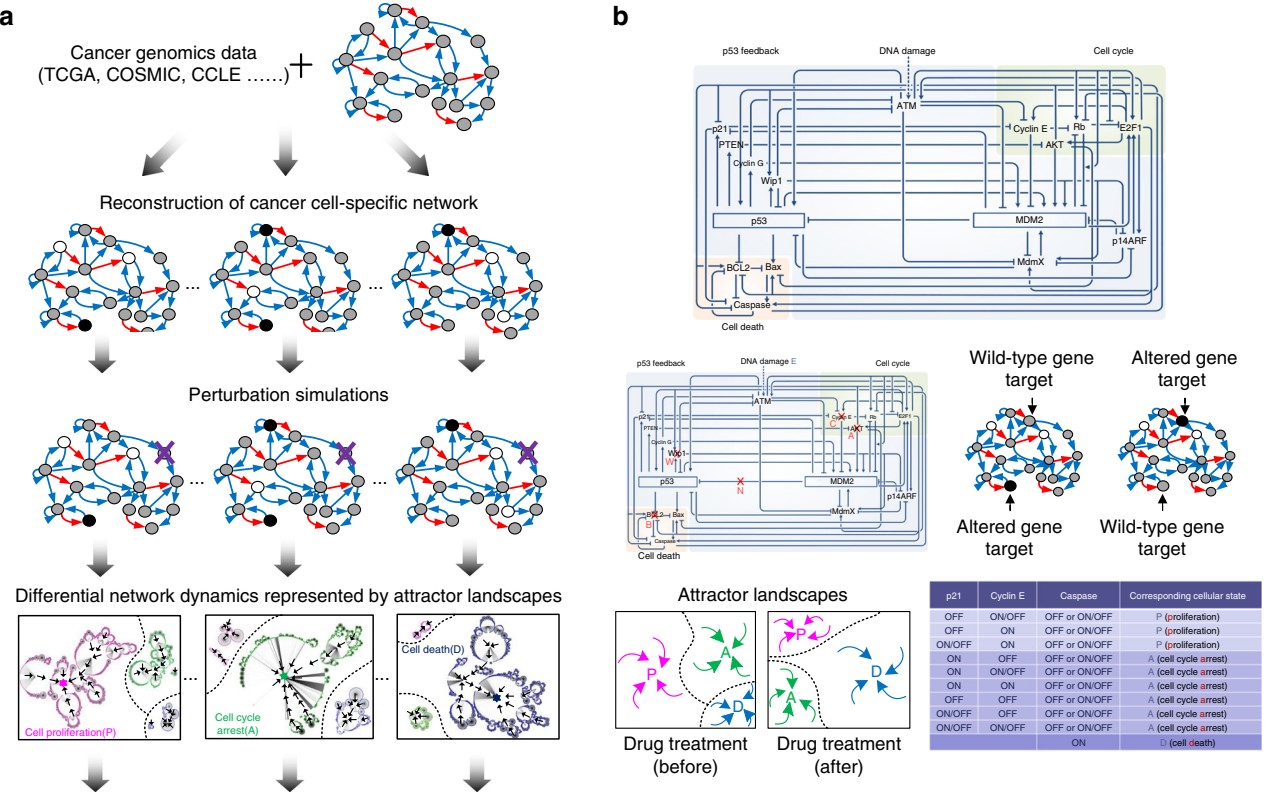

**Fig. 1** Network dynamics-based stratification of cancer cells using attractor landscape analysis of network dynamics and application to p53 network. **a** Cancer cells are represented as differentially wired networks that have a distinct network topology by mapping functional genomic alterations in cancer cells onto the nominal network. Network dynamics induced by the perturbation can be analyzed by an attractor landscape, which consists of the trajectories from all possible initial states to the attractor states. Different cancer cells eventually reach the attractor states that correspond to specific cellular phenotypes. The area around each attractor state is the region of states with trajectories going to the attractor, which is called the "basin of attraction" or "basins", and can be used for measuring the relative ratio of the specific cellular phenotypes, including cell proliferation (P), cell cycle arrest (A), and cell death (D). Final stratification of the differentially wired network is obtained based on differential network dynamics in response to perturbation. The node color represents the status of the node activity. For example, a black (white) node means that the node is constantly activated (inactivated) and a gray node means that the status is dependent on the activity of a given input. **b** The p53 network is modeled as a simplified Boolean network, consisting of 16 nodes with multiple feedback loops through p53 (upper panel). We chose to analyze p53 network in response to perturbation of five druggable network nodes/links (left, middle panel: AKT (A), BCL2 (B), Cyclin E (C), Wip1 (W), and p53-MDM2 (N)), with/without a DNA-damaging reagent, Etoposide (E). The selected target nodes were further divided into two classes: one corresponds to altered gene in the network that is constantly activated or inactivated and the other corresponds to wild-type gene (right, middle panel). To evaluate drug response variation, we defined the viable cellular phenotypes based on attractor states (lower panel)

projected the relevant genomic alteration profile onto the molecular interaction network to produce a differentially wired network, which represents a simple, but essential, genomic landscape as observed in distinct cancer cells. Next we analyzed its state transition dynamics for various perturbations that mimic the drug action by changing either node activity or interaction type. These perturbations led to changes of the attractor landscape and the relative ratios of the distinct attractor states (i.e., cellular phenotypes). Based on results of the perturbation analysis, we clustered a large panel of human cancer cell lines profiled by the CCLE project into drug response subgroups, according to their major cellular phenotypes (Fig. 1a); and we also evaluated efficacy of distinct drug (i.e., the trigger of perturbation) and synergistic effect of drug combinations for the different cancer cell types.

**Network dynamics-based analysis of the p53 network.** One of the most well-characterized genomic alterations associated with cancer is downregulation of the activity of a tumor suppressor gene, *p53*. p53 mutation is observed in about 50% of all cancers

and is believed to be a major cause of drug resistance due to loss of p53-mediated apoptotic signaling[23]. However, resistance to apoptosis is also observed in many cancer cells that have wild-type p53[24,25], indicating p53-mediated cellular response not only depends on p53 itself but also collective activities of other p53 signaling pathway components. Mechanistic understanding of how resistance may arise in cancers with wild-type p53 is very limited and thus becomes particularly important. To investigate the dynamic process of variable drug responses with respect to altered regulation of the p53 network, we applied the above network dynamics-based analysis to a p53 network model, which consists of major p53 signaling pathway components and multiple feedback loops and crosstalk between them (Fig. 1b, upper panel). Our previous analysis of this p53 network model has shown that attractor landscape analysis can identify key feedback loops in the p53 network and predict p53 network-mediated cellular response to DNA-damaging chemotherapeutics, etoposide[21].

Here we extended the p53 network perturbation analysis for more drugs and drug combinations. Highly specific small-

molecule inhibitors are currently available for one link and four nodes of the p53 network, including Nutlin-3 for p53-MDM2, GSK2830371 for Wip1, MK-2206 for AKT, CDK2 inhibitors for Cyclin E, and Navitoclax for BCL2 family proteins[26–32]. We thus chose to analyze attractor landscape resulted from inhibitory perturbation of these five targeted drugs, applied either alone or in pairs, and with or without the DNA-damaging drug, etoposide. To simulate network perturbation that mimic the mechanism of drug action, we assigned "OFF" to node or link that is the inhibitory target of the specific drug treatment in the attractor landscape analysis. That is, four nodes (AKT (A), BCL2 (B), Cyclin E (C), Wip1 (W)) and one link (p53-MDM2 (N)) in the p53 network may be "OFF", in the presence or absence of DNA-damaging drug, etoposide (E) (Fig. 1b, left, middle panel). By analyzing changes of the attractor landscape with an inactive node or link as a result of inhibitory perturbation, systemic variation in responses to drug was investigated. In addition, based on the status of each network component, the selected drug targets were further divided into two classes: one corresponds to altered gene in the network that is constantly activated or inactivated, and the other corresponds to wild-type gene in the network (Fig. 1b, right, middle panel). Targeted therapies in general focus on inhibition of the altered gene, which is an intuitive and common approach compared to that of wild-type targets. However, results of our analysis, as elaborated below, suggested that inhibition of wild-type gene also effectively alters network dynamics to the desired phenotype.

We defined drug response phenotypes based on attractor states from the distinct network dynamics so as to evaluate drug response variation (Fig. 1b, left, lower panel). Due to complexity of a biological system, it is impossible to describe all phenotypes comprehensively in any given model. We thus aimed to acquire the most appropriate level of detail in terms of phenotypes, based on prior knowledge of the relevant pathways. In the case of drug response, we consider three cellular phenotypes broadly classified as: cell proliferation (P), cell cycle arrest (A), and cell death (D), to be the most relevant. And we defined attractor states with persistent activation of Cyclin E to be the cellular phenotype of cell proliferation; attractor states that resulted in persistent activation or oscillatory activation of p21 were defined as the phenotype of cell cycle arrest; and attractor states that resulted in persistent activation of caspase were defined as the phenotype of cell death (Fig. 1b, right, lower panel, see Supplementary Fig. 1 in details). Consequently, major cellular response phenotype is determined by measuring and comparing the basin size of the attractor states that correspond to each of the three cellular phenotypes (P, A, and D), i.e., their relative ratio of occurrence, in each cancer-specific network in response to the drug-induced perturbation. We elaborated below the key computational procedures to perform cancer cell stratification based on drug response mediated by the p53 network. Details of the acquired drug response profiles of the cancer-specific p53 networks can be found in Supplementary Data 2.

**Mapping genomic alterations to network modifications**. The first step of our computational approach is to construct cancer-specific p53 networks, using genomic data from the CCLE database for p53 network components from 83 human cancer cell lines, which all have wild-type p53 and functional caspases, and represent 14 different tissue origins. The workflow is summarized in Supplementary Fig. 2. Briefly, we first annotated the thousands of genomic changes associated with p53 network and then curated a list of candidate genomic alterations that have direct functional effect on the network dynamics. We integrated copy number alterations (CNA) and somatic mutations from whole-

exome sequencing data obtained from the cBioPortal for Cancer Genomics[33,34]. To filter out genomic alterations that were unlikely functional, only missense mutations that have a high or medium functional impact score were selected, as well as truncation mutations[35]. Also, we picked out genes with CNA that have corresponding changes in messenger RNA (mRNA) expression levels. In total, we selected 191 candidate functional genomic alterations for the network analysis. These functional alterations include mutations (42 truncation mutations and 18 missense mutations) and CNA (50 HOMDEL (homozygous deletion), 11 LOSS, 28 GAIN, and 42 AMP (amplification). We think this set of functional genomic alterations provides a concise description of the genotype of the cancel cell lines with respect to p53 network function (Fig. 2a; Supplementary Data 1).

We next determined the functional outcome of each genomic alteration, i.e., whether it is gain of function or loss of function, or null. The alterations were analyzed in a binary fashion, such that an altered gene (protein) was either constantly activated (A) or constantly inactivated (I), depending on its alteration type in a given cancer cell line (Supplementary Fig. 3). For example, a particular gene with CNA, such as "AMP" and "HOMDEL", are denoted as "CNA_A" and "CNA_I", respectively. As mutations can contribute to cancer progression by activating or inactivating protein function, missense mutation is denoted as "MUT_A" or "MUT_I", based on its functional type, e.g., oncogene or tumor suppressor, as curated by OncoKB[34]; and nonsense mutation is denoted as "MUT_I". These genomic alterations are known to be not exclusive to one tumor type, nor are they, with few exceptions, present in 100% of the samples in a particular tumor type (Fig. 2b).

With all altered genes functionally annotated, we then projected the selected functional genomic alterations onto the p53 network model. The selected functional alterations were considered as endogenous modifications (or perturbation) of the network. The node status in the p53 network was denoted as either constantly activated (A), constantly inactivated (I), or input-dependent (N), based on the functional annotation results (Fig. 2c). Based on the projection results, we were able to group cancer cell lines with the same node activity profile to one single topology of p53 network. In total, we identified 45 differentially wired p53 networks (DWNs) from the 83 human cancer cell lines, which involved 1–5 network alterations (Fig. 2d). Our analysis also immediately revealed a lack of correlation between genotypes and tissue origins of the cancers, i.e.: (1) cancer cell lines originated from the same tissue origin vary substantially in node modifications, and (2) some similar node modification patterns are observed in cancer cell lines from different tissue origins. In other words, the differentially wired p53 networks demonstrated intra-cancer type heterogeneity and cross-cancer type similarity (Fig. 2c).

**Identifying drug response phenotype by network perturbation**. We next clustered the 45 differentially wired networks (DWNs) based on network similarity calculated by specific combinations (signatures) of the network components, each taking on one of the three possible statuses: A, I, or N (Fig. 3a). As a result, the 45 DWNs were clustered into three major subgroups, which have functionally unique network perturbation. DWNs belonging to subgroup 1 (denoted in green) all have constantly inactivated p14ARF. Members of subgroup 3 (yellow) all have constantly activated AKT. And we grouped the rest of the DWNs, which share no common similarity, into subgroup 2 (pink) (Fig. 3a).

To assess whether DWNs belonging to the same subgroup have similar drug response profiles, we conducted perturbation simulations for the 45 distinct network subtypes, by assigning

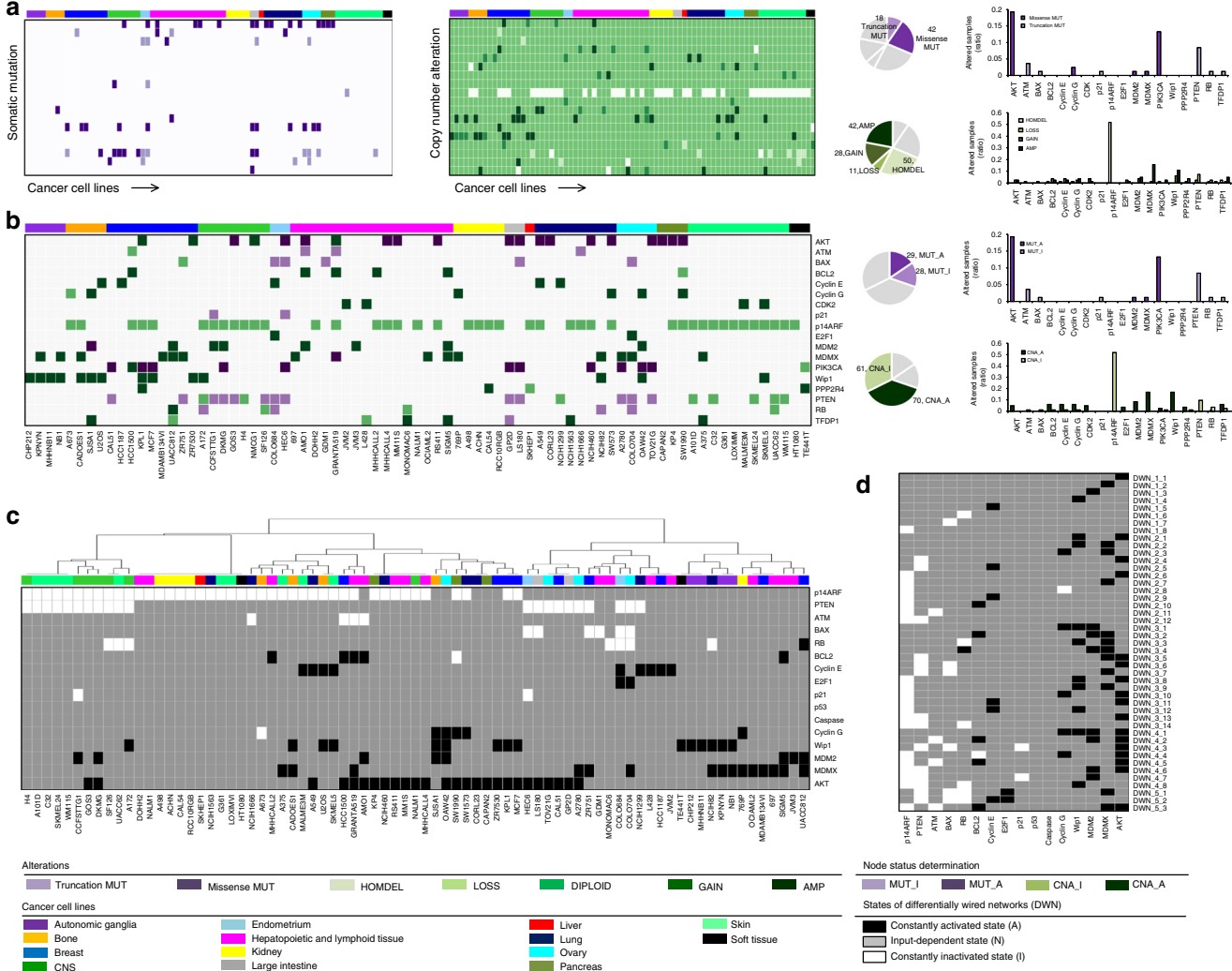

**Fig. 2** Mapping cancer-associated genomic alterations to the p53 network. **a** Genomic alterations considered in our analysis include copy number alterations and somatic mutations. To select the functional genomic alterations, we first reduced thousands of genomic alterations to a few hundred candidate functional events (heatmaps to the left). Copy number alterations (HOMDEL(homozygous deletion): white green; LOSS: light green; DIPLOID: bright green; GAIN: green; AMP(amplification): dark green, various shade of greens), and somatic mutations (truncation: light purple; missense: dark purple) define the genetic landscapes of the 83 human cancer cell lines from 14 cancer types (arranged from left to right with groups of columns labeled by cancer type). The selected alterations tend to involve well-known oncogenes and tumor suppressors (histograms) and the pie charts show the proportion selected. **b** Selected functional alterations that are either gain of function or loss of function. The selected alterations were associated with cancer cell lines in a binary fashion, such that a gene (protein) with the alteration was either constantly activated (A) or constantly inactivated (I), depending on its alteration type in a given cancer cell line (MUT_I: light purple; MUT_A: dark purple; CNA_I: light green; CNA_A: dark green). **c** Projection of the selected functional genomic alterations onto the nominal p53 network. Node status in the p53 network is determined in a ternary fashion, such that node activity is either constantly activated (A), constantly inactivated (I), or input-dependent (N) (A: black; N: gray; I: white). **d** The 45 distinct differentially wired p53 networks (DWNs) constructed based on the genomic data. Cancer cell lines that have the same node activity profile were matched to an identical single network

"OFF" to node or link that is the inhibitory target of specific drug treatment. From the perturbation simulations, systemic variation in drug response can be investigated by analyzing changes of the attractor landscape with an inactive node or link as a result of inhibitory perturbation. Based on the attractor landscape analysis, we were able to identify specific drug response phenotypes of the 45 DWNs under both single and combinatorial inhibitory perturbation. The unique strength of our network dynamics-based approach is that our results allowed us to identify major cellular phenotype for drug response and the source of variability in drug response between distinct cancer cells at the functional network level, beyond individual gene/protein.

If a drug-induced perturbation is able to change the major cellular response phenotype, we define the drug-targeted node as "critical target" for the particular network subtype (Fig. 3b). Furthermore, if an inhibitory perturbation results in different change of major cellular response phenotype in cancer cell vs. normal cell (i.e., p53 network with no genomic alteration), we considered the particular genomic alterations present in the cancer cell as the determinant of drug response for the network subtype. We were able to identify a minimal subset of such genomic alterations for a given inhibitory treatment, which we termed "critical determinant". Intuitively, if a drug-induced perturbation results in the same major cellular response phenotype in cancer and normal cell, there is no critical

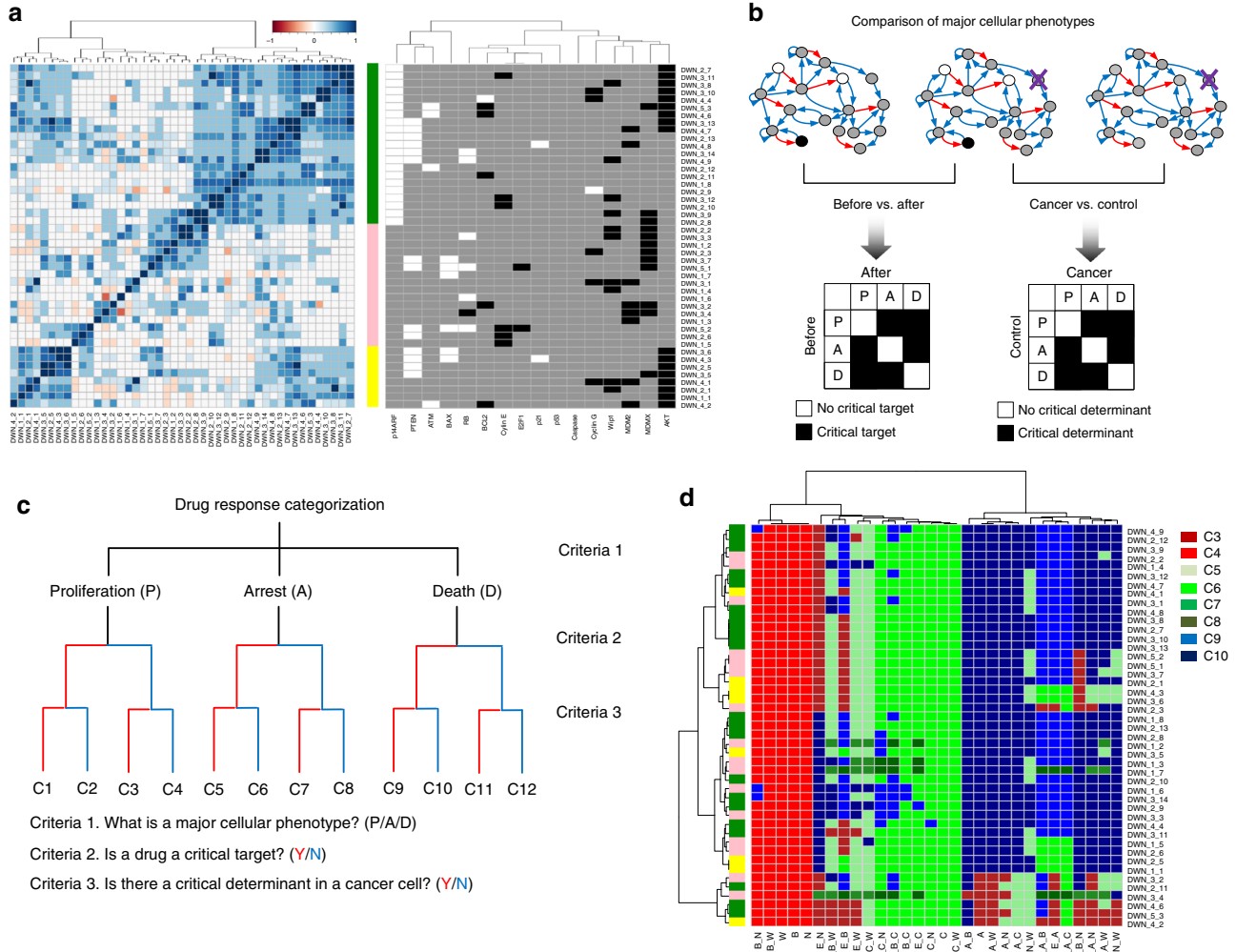

**Fig. 3** Classification of cellular response to single and combinatorial perturbations. **a** Network similarity-based clustering of 45 differentially wired networks. They were divided into three subgroups (subgroup 1: green, subgroup 2: pink, subgroup 3: yellow). **b** A major cellular response phenotype of cancer network after drug treatment is compared with that before drug treatment to identify the presence or absence of "critical target". Also, it is compared with a major cellular phenotype of control (normal) network to identify the presence or absence of "critical determinant". **c**, **d** Based on results of the attractor landscape analysis, heterogeneous drug responses of the 45 p53 network subtypes from the 83 human cancer cell lines were classified into eight distinct response groups by three criteria, including the major drug response phenotype (P/A/D), the presence or absence of critical target, and the presence or absence of the set of critical genomic determinant

determinant in the cancer cell for this drug. The whole response profiles from simulation of all possible perturbations are necessary for identifying the "critical determinant" of the various DWNs. We provided details of these profiles in Supplementary Data 3. Overall, this systemic analysis allowed us to categorize and understand the cancer-specific networks in terms of drug response based on whether the network has critical target and/or critical determinant. The critical determinants are particularly interesting, as they provide new angle to understand drug resistance mechanism and design combinatorial therapeutic strategy. Examples of drug response profiles for identifying critical target and critical determinant were provided in Supplementary Figs. 4 and 5.

The three criteria as discussed above, including the major drug response phenotype (P/A/D), the presence or absence of critical target, and the presence or absence of the set of critical genomic determinant, enabled us to classify the heterogeneous drug responses of the 45 p53 network subtypes from the 83 human cancer cell lines into eight distinct response groups (Fig. 3c, d). Interestingly, we found that clustering based on only network topology (Fig. 3a) is not sufficient to predict drug responses; and

drug responses can be distinct, even if networks have similar properties (Supplementary Fig. 6). This illustrates the importance of analyzing differential network dynamics to predict drug response. Our network dynamics-based method of stratification is thus more informative than the previous methods that only considered the static topology. Our results revealed not only the drug response phenotypes across different drug treatments and cancer cell types, but also the critical genomic determinants relevant to drug resistance. Based on this result, we were able to identify a set of critical determinant of each DWN for specific drug perturbation. This set of genomic alterations can be potentially employed as biomarkers to predict drug response and also be exploited as drug combination targets to sensitize treatment response of specific cancer subtypes. We provided the list of critical determinant that may be used as biomarkers in Supplementary Data 3.

**Predicting therapeutic response based on network dynamics.** To examine variation between cancers and drug treatments more directly, we next quantified results from the network perturbation

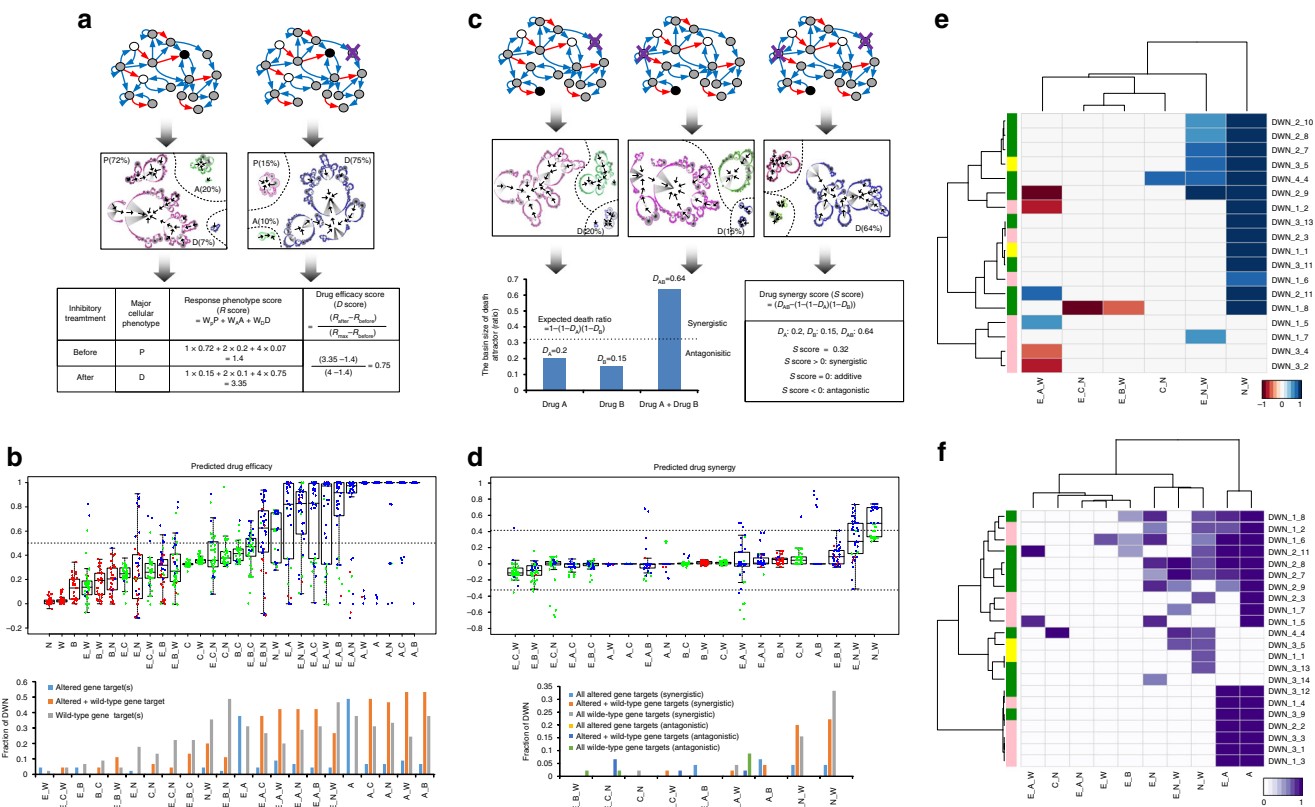

**Fig. 4** Quantitative scores of differential cellular response to perturbations. **a** An example of the predicted drug efficacy score. **b** (upper panel) Distribution of the predicted drug efficacy (D score). Each box plot shows the distribution of drug efficacy scores of 45 distinct networks. The bold lines and the boxes represent the median and the interquartile range (25th and 75th percentiles) and the whiskers extended to 1.5 times the interquartile range. If D score > 0.5, we consider the corresponding drug is effective. Dot color denotes major cellular phenotype of each differentially wired network (DWN) in response to perturbation (red: P, green: A, blue: D). (lower panel) Distribution of the effective drug targets (i.e., high drug efficacy, when death is the major cellular phenotype). Effective drug targets are divided into all altered gene target, combination of altered gene target, and wild-type gene target and all wild-type gene target. **c** An example of the predicted drug synergy score. Y-axis is the ratio of corresponding DWNs to total DWNs for the effective inhibitory treatment. **d** (upper panel) Distribution of the predicted drug synergy score (S score). S scores are converted to z-scores to facilitate comparison and definition of the synergistic (upper line)/antagonistic (bottom line) thresholds. Box plots show the median (the bold lines), the interquartile range (the boxes), and the lowest and highest scores within 1.5 times the interquartile range (the whiskers). Dot color denotes major cellular phenotype after perturbation (red: P, green: A, blue: D). (lower panel) Distribution of the selected synergistic/antagonistic drug combinations, consisting of altered gene target and wild-type gene target. Y-axis is the ratio of corresponding DWNs to total DWNs for the effective inhibitory treatment. **e** Selected synergistic or antagonistic drug pairs that comprise wild-type gene targets based on the drug synergy score. **f** Prediction of effective network-specific drug target(s) that comprise wild-type gene targets

analysis into specific scores, including response phenotype score (R score), drug efficacy score (D score), and drug synergy score (S score) (refer to Supplementary Data 4 for details). The response phenotype score was calculated as sum of the products of the basin ratio of each attractor state (i.e., proliferation (P), cell cycle arrest (A), and cell death (D)) and their assigned weighting ($2^0$ for P state, $2^1$ for A state, and $2^2$ for D state). Intuitively, if the R score is close to 1, proliferation is the major phenotype, while R score close to 4 indicates a major phenotype of cell death. Based on the response phenotype score, we calculated the drug efficacy score as follows,

$$\text{Drug efficacy score} \, (D \, \text{score}) = \frac{R \, \text{score}_{\text{after}} - R \, \text{score}_{\text{before}}}{R \, \text{score}_{\text{max}} - R \, \text{score}_{\text{before}}}$$

where R score$_{\text{after}}$ and R score$_{\text{before}}$ are the response scores with and without drug treatment, respectively, and R score$_{\text{max}}$ is 4, which is the maximal anticancer effect (i.e., all cell death) that a drug can induce (Fig. 4a).

The derived drug efficacy scores of the 45 DWNs, i.e., the p53 network subtypes, showed that DWNs have distinct drug efficacy

score and major cellular phenotype under the same inhibitory treatments. When drug efficacy score is larger than 0.5, the corresponding inhibitory perturbations mainly trigger cell death, with only a few exceptions (Fig. 4b, upper panel). We thus selected network perturbations that showed a high drug efficacy score (>0.5) and mainly induced cell death as effective drug targets. These effective drug targets include both altered genes and wild-type genes in the network (Fig. 4b, lower panel). The altered gene targets of each network acquired from our analysis are largely intuitive, as these are mostly oncogenes and thus known targets of anticancer therapy. However, the wild-type gene targets revealed by our analysis are highly novel and provide new candidate targets for cancer-specific treatment development.

Moreover, our results predicted not only effective single target but also combined targets across the 45 DWNs. Inhibition of the combined targets showed a high drug efficacy score (>0.5) and mainly induced cell death. Interestingly, most network perturbations involving AKT inhibition are effective regardless of the network subtypes, indicating AKT is an attractive therapeutic target to overcome cell-type specific drug resistance. Nonetheless, targets, such as AKT and Cyclin E, may be difficult to inhibit

completely due to their multiple functional sites and/or redundancy of the kinase signaling pathways[36]. Recent study has indeed shown that strong cell death is only induced by triple inhibition of AKT, but not by single inhibition[37–39]. In such case, one needs to look for other targets that are effective and easy to inhibit as alternative therapeutic strategy. And our results revealed some possible alternative targets, whose simultaneous inhibition could be highly effective. For instance, combinations, such as inhibition of p53-MDM2 and Wip1 with or without DNA-damaging drug, inhibition of p53-MDM2 and BCL2 with DNA-damaging drug (N_W, E_N_W, E_B_N), all showed effect in triggering cell death in more than 50% of the DWNs. Hence, systemic computational analysis of cancer cell response, such as our network dynamics-based analysis, can be particularly informative in terms of identifying non-intuitive, wild-type gene targets for developing new treatment strategy.

To further zero in on drug combinations that are synergistic, we calculated a drug synergy score (S score), which is defined as follows[40],

$$\text{Drug synergy score } (S \text{ score}) = D_{AB} - D_A - D_B + D_A \cdot D_B$$

where $D_{AB}$, $D_A$, and $D_B$ are the extent of cell death induced by combinatorial treatment and individual treatment, respectively (Fig. 4c). Intuitively, if two inhibitory treatments act independently and do not induce synergistic effect when combined, S equals to 0. And S score > 0 indicates synergy and S score < 0 indicates antagonism (Fig. 4d). We calculated S scores for a total of 1000 drug pairs and plotted the distribution, which follows a normal distribution (Supplementary Fig. 7). We subsequently converted the distribution to z-scores to facilitate comparison and selection of synergistic/antagonistic thresholds. Based on the distribution, we defined drug pairs that have z-scores of <−1.645 or >1.645 (corresponding to the 5th and 95th percentile of a normal distribution, respectively) as having significant synergistic or antagonistic effects (Fig. 4d, upper panel). Results of this synergism analysis showed that although drug combinations involving AKT triggered strong cell death (i.e., major cellular response phenotype is D), the combined effects are mostly additive (S score is close to zero). On the other hand, we found that combination of inhibition of p53-MDM2 and Wip1 (i.e., N_W, E_N_W) exhibited the strongest synergistic effect in activating cell death, regardless of network subtypes (Fig. 4d, lower panel). Figure 4e shows synergistic or antagonistic drug pairs that consist of wild-type gene targets based on the drug synergy scores. This result again illustrates that the network dynamics-based approach can not only identify network-specific effective drug pairs but also reveal wild-type gene targets for novel drug combinations.

In Fig. 4f, we summarized the effective drug(s) that targets wild-type gene and drug combination(s) for the cancer-associated p53 network subtypes. The effective wild-type gene targets share three common properties in changing the network dynamics: (1) they induced mainly cell death, (2) the resulting drug efficacy score is high (D score > 0.5), and (3) strong synergistic effects arise when they are combined. The effective wild-type gene targets clearly depend on the network subtypes. Overall, network dynamics-based analysis of the p53 network demonstrated the effectiveness of our approach to computationally investigate therapeutic strategy across a large number of cancer types that have been genotyped. And such systematic stratification of cancers, based on network dynamics induced by targeting therapeutically actionable network alterations, provides a potentially useful method to quantitatively predict drug combination that can reduce cell-type-specific response variation as well as design more effective combinatorial treatment strategy.

**Experimental validation of network dynamics-based analysis.** To validate our method in predicting anticancer drug response in cancer cell lines with distinct genetic backgrounds, we compared the predicted drug response profiles with experimental results of eight distinct cancer cell lines in the panel, including A549 (derived from lung cancer), MCF7 and CAL51 (derived from breast cancer), U2OS and SJSA1 (derived from bone cancer), A2780 (derived from ovarian cancer), A375 (derived from skin cancer), and 769P (derive from kidney cancer). Based on their specific genomic alterations, these cancer cell lines (A549, MCF7, CAL51, U2OS, A375, A2780, and 769P) are matched to the p53 network subtypes of DWN_3_11, DWN_3_8, DWN_2_4, DWN_3_12, DWN_3_5, DWN_2_7, DWN_3_5, DWN_2_3, respectively. Moreover, in terms of network similarity, A549, MCF7, U2OS, and A375 belong to subgroup 1 (denoted in green in Fig. 3a); 769P and SJSA1 belong to subgroup 2 (denoted in pink); and A2780 and CAL51 belong to subgroup 3 (denoted in yellow) (Fig. 5a). However, as discussed above, clustering based on network topology alone does not implicate DWNs in the same subgroup have the same drug response profiles. In Fig. 5b, we investigated the critical determinant for each drug treatment in the eight distinct cancer cell lines. Clearly, network subtypes in the same topology subgroup did not show the same response phenotype. For instance, inhibition of BCL2 (B) with DNA-damaging drug (E) resulted in a major response phenotype of proliferation in A549 and MCF7, but a major response phenotype of cell death in A375 and U2OS. Furthermore, although A549 and MCF7 share two alterations (constantly inactivated p14ARF and constantly activated AKT), their critical determinant was different: Cyclin E(A) for A549, while p14ARF(I) and AKT(A) or Wip1(A) and AKT(A) for MCF7. This result again demonstrates that the variable cancer cell-specific drug responses are determined by differential p53 network dynamics, not simply their network topology.

We chose to particularly look at effect of three inhibitory treatments experimentally, i.e., BCL2 (B), Wip1 (W), and p53-MDM2 (N), and their combinations, as BCL2 is the common wild-type gene target in the eight distinct cancer cell lines, and Wip1 and p53-MDM2 are frequently altered gene targets. (Fig. 5a; Supplementary Data 4). For validation, we focused on the cell death phenotype, as cell death is the key therapeutic response where the distinct cancer types vary the most. Detailed results from simulation and experimental measurements were provided in Supplementary Figs 8–10 and Supplementary Data 5. To evaluate the agreement between model predictions and experimental results, we calculated the Pearson correlation coefficient as well as the root mean square error (RMSE) for drug responses of the eight cell lines (Fig. 5c). Intuitively, a higher Pearson correlation coefficient with the experimental results and a lower RMSE indicate better performance of the network dynamics-based approach in predicting drug response of a given cell line-specific network. The overall Pearson correlation across all the cell lines combined is high (correlation coefficient: 0.75, p < 0.001), indicating that the drug responses are well predicted by our network dynamics analysis. In addition, Fig. 5d plotted values of 1/RMSE with respect to the Pearson correlation coefficients for each cell line, illustrating that our method performs quite well in both statistical measures, in particular for cell lines at the upper right-hand corner of the graph.

However, we do note significant discrepancy between some modeling and experimental results. For example, a relatively high RMSE of CAL51 was observed as compared to the other seven cancer cell lines, even though the Pearson correlation of CAL51 was high. For CAL51, the experimentally observed cell death responses were larger than the predicted responses under all drug treatments, including the DNA-damaging drug alone (E). We thus suspect the discrepancy may arise from deficiency of the CAL51

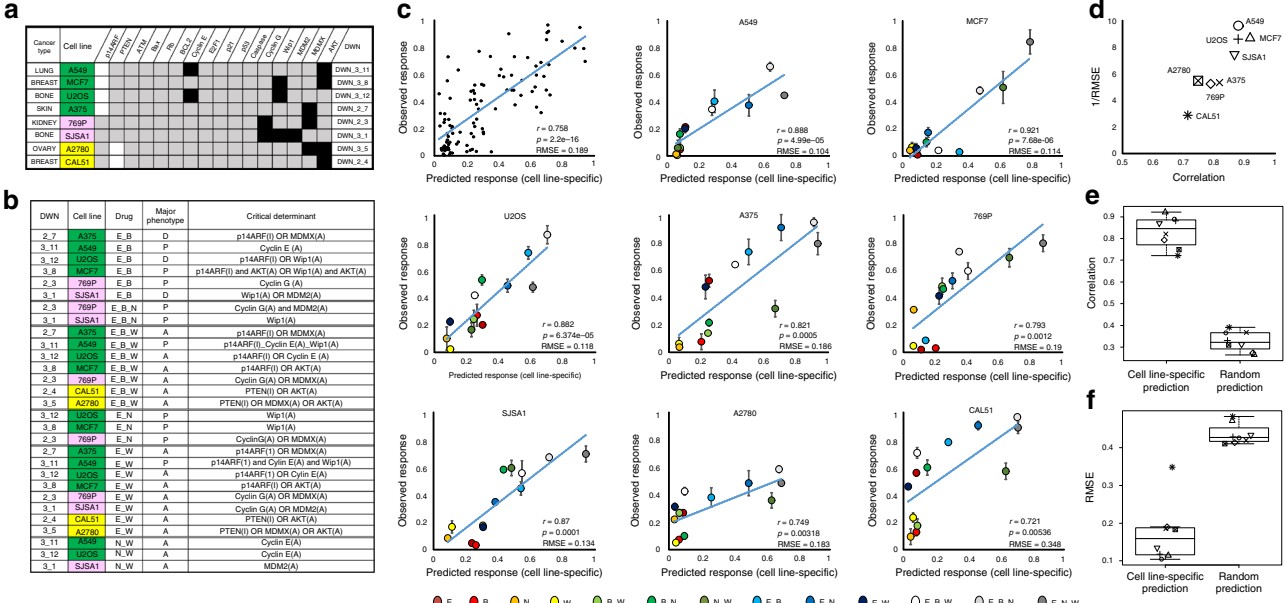

**Fig. 5** Experimental validation of the drug response profiles of eight cancer cell lines in the panel. **a** Differentially wired networks corresponding to the eight distinct cancer cell lines (A549 (lung, DWN_3_11), MCF7 (breast, DWN_3_8), U2OS (bone, DWN_3_12), A375 (skin, dwn_2_7), 769p (kidney, DWN_2_3), SJSA1 (bone, DWN_3_1), CAL51 (breast, DWN_2_4), A2780 (ovary, DWN_3_5)). **b** Predicted major drug response phenotype and the associated critical determinant for the eight cell line-specific networks in response to various perturbations. **c** Scatter plots of the experimentally observed vs. predicted cell death ratio for all cell lines combined and each individual cell line. The error bars indicate the standard deviation of time-lapse imaging experiments under various drug treatment conditions. **d** Evaluation of drug response prediction results acquired by our network dynamics analysis using Pearson correlation and RMSE. **e, f** Comparison of the predictive power of our network dynamics analysis and random prediction using Pearson correlation and RMSE

network that we curated due to: (1) incomplete information from the genomic data that limits the accuracy of network mapping and subsequent network simulation analysis; and/or (2) the use of a simplified p53 network in our study, where genomic alterations of additional p53 network components that are not included in our network model may play an important role in regulating drug responses. For instance, our CAL51 network model does not include some critical genomic alterations observed in CAL51, such as mutations of *MAP2K4* and *CHEK1*. *MAP2K4* is known to activate JUN kinase and p38 signaling pathways, and is involved in p53-mediated response, such as cell cycle arrest and cell death[41]. *CHEK1* is also known to coordinate cell cycle checkpoint response and DNA damage response. Mutation of *CHEK1* could lead to the loss of CHK1, resulting in increased sensitivity to DNA-damaging agents, such as etoposide[42].

To further examine the predictive power of our network dynamics analysis, we compared our cell line-specific predictions with random predictions acquired by shuffling alterations of each cell line such that the number of alterations is preserved, while their locations are randomized (Fig. 5e, f). For all random predictions, we observed relatively weak correlations between the experimentally observed and randomly predicted responses, compared to that between the experimental data and predicted responses by our approach ($p < 0.001$, Wilcoxon rank sum test). In addition, each RMSE of our cell line-specific predictions is significantly smaller than that of the random prediction ($p < 0.001$, Wilcoxon rank sum test). Overall, these results demonstrate that our network dynamics analysis performs substantially better in predicting drug response than random prediction.

## Discussion

Previous computational approaches to predict drug response phenotype from cancer genotype in general involve using statistical models that project genomic alteration profiles onto a

molecular interaction network, without explicitly considering activation/inhibition (Fig. 6a). Despite progress made by these statistical approaches, our knowledge and mechanistic understanding of cancer heterogeneity and its impact on variable drug responses remain limited. In this study, we proposed a new method to analyze the existing cancer genomics data by considering dynamic response of a specific molecular network crucial for mediating drug response. We used the p53 network as an example to illustrate the effectiveness of our method in elucidating how genomic alterations in cancer cells rewire the topology of a signaling network and thereby change its dynamics upon stimulation, such as pharmacological perturbations. This network dynamics-based approach allowed us to not only stratify cancer cells in terms of functional dynamics but also predict cell-specific drug responses. As our study demonstrated that network dynamics, rather than network topology, determine distinct drug responses, stratification based on network dynamics is likely capable of better predicting clinical outcome than previous methods.

A key contribution of our study is that we developed a method to functionalize genomic data onto dynamic response of a signaling network based on attractor landscape analysis and were able to categorize network response profile to distinct perturbations for a large panel of cancer cell lines. Our analysis results clearly show that individual components of the network or network topology alone are not sufficient to predict drug responses. Members of the same network subgroups clustered by common network characteristics often exhibit different responses to the same drug perturbation. This suggests that collective alterations in the signaling network have to be considered for evaluating efficacy of the drugs and designing biomarkers to predict drug response. One interesting finding from our analysis is the sets of "critical determinant" for different cancer network subtypes that can effectively determine drug responses and thus may be

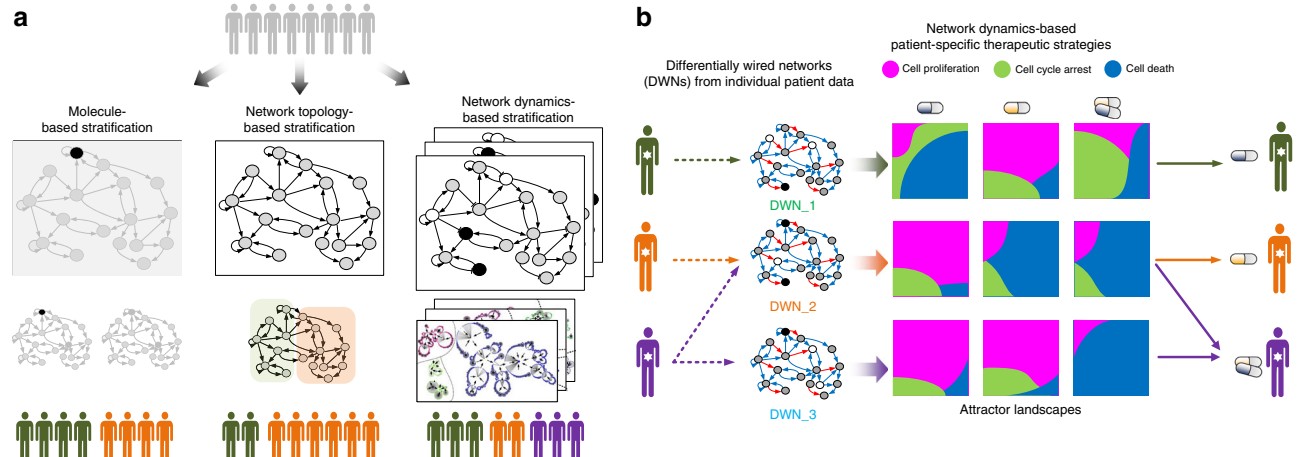

**Fig. 6** Network dynamics-based therapy. **a** Various approaches to stratify cancer patients based on their genomic profiles. **b** The workflow of employing network dynamics-based analysis to design patient-specific therapeutic strategies

employed as collective biomarkers and/or targets for combinatorial treatment. For instance, for drugs that mostly induce proliferation or cell cycle arrest when used alone, one may combine them with inhibitors of their critical determinants, to activate strong cell death.

In this study we used a simplified p53 network model for the attractor landscape analysis, which clearly has its limitation. The network analysis cannot be applied to cancer cells with mutant p53; and the simplified model may miss p53 network components, whose genomic alterations could play an important role in regulating p53 pathway-mediated drug response. Moreover, as our study is based on Boolean modeling that discretized genomic alterations as either on or off, our method cannot distinguish between weakly and strongly activating mutations or consider varying efficiencies of the inhibitors. Nonetheless, we think our approach provides a foundational framework that can be further developed to address the above limitations. For instance, we can easily expand the simplified p53 network to a larger, more comprehensive functional network that incorporates additional components crucial to oncogenesis, metastasis, and/or tumor response. We could also expand the Boolean network model to multi-valued logical model or use fuzzy logic instead of Boolean logic to describe varying degrees of activation/inhibition and drug efficacy[43,44]. As for cancer cells with mutant p53, the drug effect is likely mediated by alternative signaling pathway so the network dynamics-based analysis should be developed and performed beyond the p53 network. As more quantitative genomic data become available for diseases and disease-associated cellular processes, we think our approach, which is capable to capture more precise phenotype from genotype, is highly adaptable for different systems and diseases to investigate effects of genomic alterations on response to disease treatment and identify appropriate, patient-specific drug treatment (Fig. 6b).

## Methods

**Curating the functional genomic data and mapping to p53 network.** DNA copy number, somatic mutation, and mRNA expression data were analyzed for all cancer cell lines from CCLE. We selected 83 human cancer cell lines from 14 distinct tissue origins and these cell lines all have wild-type p53 and caspases. We consider all available genomic data types in our analysis, including genome-wide DNA copy number information, and mutation data for genes associated with the p53 pathway. Mutation frequencies were calculated as the ratio of mutation counts to number of bases covered. To focus on mutations most likely to be functional, mutations in introns, untranslated regions, flanking, and intergenic regions, as well as silent and RNA mutations, were excluded. The CCLE database provides the number of reads per base in the sequenced regions, so the number of bases covered was given by the number of positions with one or more reads. To filter out events that were likely non-functional, only genes with CNA that have concordant

changes in mRNA expression levels, when compared to wild-type cases, were selected. In total, we curated 191 candidate functional alterations. These alterations were considered in a binary fashion, such that an alteration either occurred or did not occur in a given cancer cell line. The resulting set of functional genomic alterations thus provides a concise genomic description of the cancel cell lines.

**Differentially wired networks and defining cellular phenotypes.** Functional genomic alterations were projected onto the nominal p53 network. Node status of the p53 network was determined based on the genomic data, and assigned in a ternary fashion, such that node activity is either constantly activated (A), constantly inactivated (I), or input-dependent (N). Through this mapping process, 45 differentially wired p53 networks (DWNs) were constructed from the 83 human cancer cell lines. They each include one perturbation to four perturbations. The DWNs were denoted by the number of perturbations that they have. For example, "DWN_3_1" represents one of the differentially wired networks that have three perturbations. For cancer cell lines that have the same node activity profiles, they are mapped to an identical single network.

For the 45 distinct network subtypes, we analyzed their state transition dynamics for various anticancer drug treatments. First, we defined the cellular states known as "attractors" in the attractor landscape. Considering anticancer drug effects are mainly in cell growth and cell death, we chose to define the "attractor" cellular states as P, A, and D. In the view of attractor landscape, each attractor indicates one of the three defined cellular states.

**Boolean network modeling of the p53 network.** A p53 network model was taken from an updated version of that in our previous study[21]. It is a simplified Boolean network model consisting of 16 nodes with multiple feedback loops through p53 for analyzing the p53 network dynamics and predicting cellular response to DNA damage. With the simplified p53 regulatory network, we modeled the network dynamics using a deterministic Boolean network with a set of state transition logics defined on the basis of biological evidence. In the Boolean network model, each node is associated with a logic table that determines the output node for a given input. Network dynamics were modeled by updating the Boolean functions, triggering system transits from the initial state to the final state, in which a network state is a collective binary representation of all variables. The state of each node can be either ON (1) or OFF (0) at each time step. To compute the network dynamics, we transformed the state transition logic into a weighed sum logic with the weight and of each link and the basal level of each node. There are multiple sets of interaction weights and basal levels for the weighted sum logic of each node that satisfy the same transition logic. Among all the possible parameter sets for weights and basal levels, we chose the minimal integer values for our study. For perturbation simulation, targeted inhibition of specific node or link is reflected in the network model by assigning the corresponding node or link to be constantly "0". More details on the state transition logic together with the interaction weights and basal levels are provided in Supplementary Note 1 and Supplementary Data 6.

**Response phenotype score for cellular response to perturbations.** Using cellular states as defined above, distinct attractors in the attractor landscape under specific perturbation were assigned to a cellular state. The overall cellular response to specific perturbation is measured as the sum of products that multiply the basin ratio of attractors belonging to same cellular state and the distinct weight corresponding to the specific cellular states ($W_P$:$2^0$, $W_A$:$2^1$, and $W_D$:$2^2$). Therefore, response phenotype score was defined as follows:

$$\text{Response phenotype score } (R \text{ score}) = P \cdot W_P + A \cdot W_A + D \cdot W_D,$$

where $P$ is the basin ratio of cell proliferation attractor, $A$ is the basin ratio of cell cycle arrest, and $D$ is the basin ratio of cell death attractor. The response phenotype score ranges from 1 to 4 and is used to estimate the drug sensitivity.

**Drug efficacy score for cellular response to perturbations**. Drug efficacy score is calculated as the difference in cellular phenotype score before and after drug treatment, normalized by the difference between maximum cellular phenotype score and the phenotype score before drug treatment:

$$\text{Drug efficacy score} (D\,\text{score}) = \frac{R\,\text{score}_{\text{after}} - R\,\text{score}_{\text{before}}}{R\,\text{score}_{\text{max}} - R\,\text{score}_{\text{before}}}.$$

**Drug synergy score for cellular response to perturbations**

To evaluate synergistic and antagonist effects of combined perturbations, we employed a synergy score as follows[40]:

**Drug synergy score** $(S\,\text{score}) = \textbf{observed combined effect} - \textbf{expected additive effect}$,

**observed combined effect** $= D_{AB}$,

**expected additive effect** $= 1 - (1 - D_A) \cdot (1 - D_B)$,

$S\,\text{score} > 0 : \textbf{synergistic}, S\,\text{score} < 0 : \textbf{antagonistic}, \text{and } S\,\text{score} = 0 : \textbf{additive}$

where $D_{AB}$, $D_A$, and $D_B$ denote the ratio of cell death induced by drug A plus B, and the ratio of cell death induced by each drug, respectively. The observed combination effect, expressed as a probability ($0 \le D_{AB} \le 1$), can be compared to the expected additive effect for probabilistic independence, i.e., $D_A + D_B - D_A \cdot D_B$, where $0 \le D_A \le 1$ and $0 \le D_B \le 1$. This multiplicative model and formula are widely used in gene knockout studies to score quantitative genetic interactions between gene deletions[45,46]. A deviation of $S$ from zero provides evidence for a non-additive interaction between the two perturbations, where $S > 0$ indicates synergy and $S < 0$ indicates antagonism.

**Experimental measurement of drug response in eight cancer cell lines**. All cell lines were purchased from American Type Culture Collection (ATCC, USA) and cultured under 37 °C and 5% $CO_2$ in appropriate medium supplemented with 10% fetal calf serum (FCS), 100 U/ml penicillin, and 100 g/ml streptomycin. MCF7, A2780, SJSA1, and 769P were maintained in RPMI; A375 and CAL51 were maintained in DMEM; U-2 OS was maintained in McCoy's; and A549 was maintained in F-12K. Etoposide was purchased from Sigma, Nutlin-3 from Tocris, and Navitoclax from Selleck. For all drug treatment experiments, Etoposide was used at 10 μM, Nutlin at 10 μM, and Navitoclax at 0.5 μM. Small interfering RNA (siRNA) for knocking down Wip1 (UUG GCC UUG UGC CUA CUA A) was custom synthesized by Dharmacon and used at 40 nM. Dharmacon On-Target plus siControl (#D-001810-01) was used as non-targeting siRNA control. siRNA transfections were performed using Hiperfect (Qiagen) according to manufacturers' instructions, and experiments were conducted 36 h after gene silencing.

To quantify the percentage of cell death induced by drug/RNAi treatment, we treated cells with single or combined drug/RNAi and then imaged the cells by time-lapse microscopy. For the imaging experiments, cells were plated in 24-well imaging plate (Cellvis, USA) and cultured in phenol red-free $CO_2$-independent medium (Invitrogen) supplemented with 10% FCS, 100 U/ml penicillin, and 100 g/ml streptomycin. Cell images were acquired with the Nikon TE2000-PFS inverted microscope enclosed in a humidified chamber maintained at 37 °C. Cells were imaged every 20 min for 48 h using a motorized stage and a ×10 objective. Images were viewed and analyzed using the MetaMorph software (Molecular Dynamics). Based on phase-contrast images of the cells, we scored cell death by cell blebbing and lysis. Percentage of cell death was calculated by normalizing the number of dead cells to the total cell number at time 0. Data were averaged from two independent imaging experiments and the total number of cells analyzed ranges from 66 to 256, varied between conditions and cell lines.

**Statistical analysis of model predictions vs. experimental data**. We quantified the difference between model predictions and experimental measurements, using the Pearson correlation coefficient and the RMSE for drug responses of each cell line. The RMSE between the observed and predicted drug responses for 13 drug treatment conditions in a given cell line is defined as follows:

$$\text{RMSE}(C, O, P) = \sqrt{\frac{1}{n} \sum_{i=1}^{n} (O_i - P_i)^2}$$

where $n$ is the number of total drug treatment conditions, $O_i$ is the observed cell death ratio of a given cell line $C$ for a treatment $i$, and $P_i$ is the predicted cell death ratio of a given cell line $C$ for a treatment $i$.

We further compared our cell line-specific predictions with random predictions acquired by shuffling alterations of each cell line such that the number of alterations is preserved, while their locations are randomized. Briefly, 480 and 4480 random networks, which are all possible networks that have two node or three node alterations, were generated by shuffling the alterations of A375, 769P, CAL51 which have two node alterations and A549, MCF7, U2OS, SJSA1, and A2780, which have three node alterations, respectively. The RMSE between the experimentally observed and randomly predicted drug responses for the 13 drug

treatment conditions in a given cell line is defined as follows:

$$\text{RMSE}(C, O, R) = \frac{1}{m} \sum_{j=1}^{m} \sqrt{\frac{1}{n} \sum_{i=1}^{n} (O_i - R_{i,j})^2}$$

where $n$ is the number of total drug treatment conditions, $m$ is the number of random networks, $O_i$ is the observed cell death ratio of a given cell line $C$ for a treatment $i$, $R_{i,j}$ is the predicted cell death ratio of random network $j$ for a treatment $i$.

**Code availability**. All codes are available from the authors upon request.

**Data availability**. All relevant data are available from the authors upon request.

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

## Acknowledgements

This work was supported by the National Research Foundation of Korea (NRF) grants funded by the Korea Government, the Ministry of Science, ICT & Future Planning (2017R1A2A1A17069642 and 2015M3A9A7067220) to K.-H.C., and Hong Kong Research Grant Council (#N_HKBU215/13 and #T12-710/16-R) to J.S.

## Author contributions

K.-H.C. designed the project; K.-H.C. and J.S. supervised the research; M.C. and K.-H.C. performed the modeling and analysis; J.S., Y.Z. and R.Y. performed the experiments; and M.C., J.S. and K.-H.C. wrote the manuscript
