## [Peer Review File · Nature Communications]

Reviewers' comments:

Reviewer #1 (Remarks to the Author):

I like the idea of the paper. The authors use attractor landscape analysis to model the dynamic network alterations. This is pretty much in line with what is emerging; namely that static GWAS does not deliver in cancer studies.

The computational analysis is *in silico*, based on publicly available datasets. They experimentally validate some key findings in cell lines as well. This part could be expanded.

The manuscript is poorly written - it needs a lot more work to make it easy to read and assess; In particular the authors tend to repeat themselves quite a bit.

Thus the paper should be completely rewritten.

But globally I don't have any major concern at this point in terms of the scientific impact.

Reviewer #2 (Remarks to the Author):

In the present manuscript, Choi et al. use the p53 network as a paradigm to study how genomic alterations in cancer cells rewire the topology of a signaling network and thereby change its dynamics upon stimulation as well as the response to pharmacological perturbations. It represents an extension of their previous work investigating the p53 network using attractor landscape analysis (Choi et al. 2012 Science Signaling). In this previous study, the authors established a boolean model that allows them to predict different states of the p53 network and associated phenotypes (proliferation, cell cycle arrest or cell death) in normal cells in presence or absence of DNA damage. This model is then adapted to represent known alterations in the cancer cell line MCF7 and to predict the effect of combination treatments such as DNA damage (etoposide) and feedback inhibition (Mdm2 inhibitor Nutlin and Wip1 siRNA).

The authors now use publicly available genome-wide datasets of somatic mutations and copy number variations to determine genomic alterations within the p53 network for 83 cancer cell lines (including MCF7) that have wild-type p53 and functional caspases. The identified alterations were discretised to represent either constantly activated or inactivated nodes and summarized in 45 differentially wired network. Next, the authors systematically investigated how the rewired networks respond to perturbation of five drugable network nodes / links in the presence or absence of DNA damage, focusing on network states that represent the phenotypes mentioned above. Using a drug synergy score, they discriminate between additive, synergistic and antagonistic combination treatments. Finally, they validate model predictions in three cell lines (U-2 OS, MCF7, A549) by determining cell death upon combinations of BCL2, Mdm2-p53 and Wip1 inhibition with DNA damage induction.

The authors present an interesting approach to systematically address how genomic alterations in cancer cells affect the function of signaling networks and contribute to the heterogeneity of the disease. The mapping and classification of different mutation types is a valuable resource for the field. However, the paper is not easily accessible. The authors introduce many new terms such as "non-intuitive target" and "critical determinant" that are not commonly used, not intuitive and not well explained. Moreover, the authors seem to rely on the reader being familiar with their previous publication. I would have appreciated a more detailed description of modelling and analysis methods to be able to understand how data is generated and conclusions are reached.

At the same time, the study seems like a direct extension from previous work, were the authors already generated a cancer cell specific network (MCF7) and predicted and validated drug combinations (Etoposide, Nutlin, Wip1 - kd). Experimental validation of new predictions is rather limited. They test two additional cell lines compared to the previous publication and add a Bcl2 inhibitor. The validated cell lines share three alterations, which means that a large space of alterations, e.g. cell lines with wild type p14/Arf, remains unexplored. A more systematic validation would be needed to complement the systematic model analysis. Moreover, the phenotypic read out relies on counting dead cells at 48h by microscopy. No errors bars are provided and no indication of cell numbers or how often experiments were repeated can be found. Here the authors should use additional quantitative methods with high statistical power for example based on flow cytometry or time-lapse imaging. It would also be helpful if the deviation of model predictions and experimental data would be quantified and put into context.

There are also some restrictions inherent to the approach of the authors. As they focus on an abstracted p53 network in p53 wt cells, alterations that are not part of the initial network and could represent truly "non-intuitive" targets are neglected. Also, p53 mutations that often acquire new functions, are also not included in the analysis. The use of a boolean network together with discretised genomic alterations limit the predictive power. For example, the authors can't distinguish between weakly and strongly activating mutations or consider varying efficiencies of inhibitors. While these limitations are inherent to the approach, the authors should at least discuss them openly.

Minor points:

- * Figure legends are extremely brief and lack detail. There are many small issues with the figures:
- * Figure 1: no explanation of e.g. node color after mapping of alterations is given, panel b is not described at all.
- * Figure 2: abbreviations such as HOMDEL, AMP etc should be spelled out at least once; heat maps in panel a should contain labels and there are typos in y-axis label for the bar graph next to pie charts ("Altered Smamples").
- * Figure 3: there are no numbers on scale for heat map
- * The naming switches between genes and nodes, e.g. CDKN2A - p14ARF, PPM1D - Wip1 in Figure 2. Here a consistent labelling would be easier for the reader.

Response to the Reviewers' Comments and Summary of Changes

Manuscript ID: NCOMMS-16-30137

Title: Network dynamics-based stratification of cancer panel for systemic prediction of anticancer drug response

Authors: Minsoo Choi, Jue Shi, Yanting Zhu, Ruizhen Yang and Kwang-Hyun Cho

Response to the specific comments of Reviewer 1:

[COMMENT #1].

“I like the idea of the paper. The authors use attractor landscape analysis to model the dynamic network alterations. This is pretty much in line with what is emerging; namely that static GWAS does not deliver in cancer studies.

The computational analysis is in silico, based on publicly available datasets.

They experimentally validate some key findings in cell lines as well. This part could be expanded.

The manuscript is poorly written - it needs a lot more work to make it easy to read and assess; In particular the authors tend to repeat themselves quite a bit.

Thus the paper should be completely rewritten.

But globally I don't have any major concern at this point in terms of the scientific impact.”

[RESPONSE] We thank the reviewer for the very positive comments regarding the novelty and significance of our work. In light of the reviewer's comment, we have restructured and completely rewritten large sections of the manuscript, in particular the Results and Discussion section, to clarify the methods of our network dynamics-based analysis and points/terms that were difficult to understand in the original manuscript. For instance, we added an overview of the modeling framework and computational procedures in the beginning of the Results section to illustrate how data were generated and results were analyzed. We also removed repeated sentences and ambiguous terms throughout the manuscript. Moreover, we extended the experimental validation from 3 to 8 cancer cell lines that now encompass all three network subgroups in terms of network topology. In general, the simulation and experimental results showed similar drug response characteristics, indicating the effectiveness of our modeling approach to predict drug response profiles from genomic data. We hope the very significant revision of the manuscript and addition of new experimental data now makes the manuscript up-to-the-standard for the reviewer to assess.

Response to the specific comments of Reviewer 2

[COMMENT #1].

“The authors present an interesting approach to systematically address how genomic alterations in cancer cells affect the function of signaling networks and contribute to the heterogeneity of the disease. The mapping and classification of different mutation types is a valuable resource for the field. However, the paper is not easily accessible. The authors introduce many new terms such as “non-intuitive target” and “critical determinant” that are not commonly used, not intuitive and not well explained. Moreover, the authors seem to rely on the reader being familiar with their previous publication. I would have appreciated a more detailed description of modelling and analysis methods to be able to understand how data is generated and conclusions are reached.”

[RESPONSE] We again appreciate the reviewer’s very positive comment on the significance and general interest of our work. Following the reviewer’s helpful suggestions, we have rewritten the manuscript comprehensively, for instance, either removing the unclear terms, such as “non-intuitive target”, or adding more detailed description to clarify the quantitative and biological meaning of the terms, such as “critical determinant” (refer to page 12 in the revised manuscript). Briefly, if an inhibitory perturbation results in different change of major cellular response phenotype in cancer cell vs. normal cell (i.e., p53 network with no genomic alteration), we considered the particular genomic alterations present in the cancer cell as the determinant of drug response for the network subtype. We then were able to identify a minimal subset of such genomic alterations for a given inhibitory treatment, which we termed “critical determinant”. Intuitively, if a drug-induced perturbation results in the same major cellular response phenotype in cancer and normal cell, there is no critical determinant in the cancer cell for this drug treatment.

[COMMENT #2].

“At the same time, the study seems like a direct extension from previous work, were the authors already generated a cancer cell specific network (MCF7) and predicted and validated drug combinations (Etoposide, Nutlin, Wip1 - kd). Experimental validation of new predictions is rather limited. They test two additional cell lines compared to the previous publication and add a Bcl2 inhibitor. The validated cell lines share three alterations, which means that a large space of alterations, e.g. cell lines with wild type p14/Arf, remains unexplored. A more systematic validation would be needed to complement the systematic

model analysis. Moreover, the phenotypic read out relies on counting dead cells at 48h by microscopy. No errors bars are provided and no indication of cell numbers or how often experiments were repeated can be found. Here the authors should use additional quantitative methods with high statistical power for example based on flow cytometry or time-lapse imaging. It would also be helpful if the deviation of model predictions and experimental data would be quantified and put into context.”

[RESPONSE] In the revised manuscript, we extended the experimental validation from 3 to 8 cancer cell lines that now encompass all three network subgroups based on network topology. Among these 8 cell lines, 4 belong to network subgroup 1 with constantly inactivated p14ARF (U-2 OS, A549, MCF7 and A375), 2 belong to network subgroup 2 (769P and SJSA1), and 2 belong to network subgroup 3 with constantly activated AKT1 (A2780 and CAL51). The 4 cell lines in subgroup 2 and 3 have wild type p14ARF. We also took advice from the reviewer to use time-lapse imaging to quantify cell death response to the single and combined drug/RNAi treatment at the single cell level. Based on phase-contrast images, we scored cell death by cell blebbing and lysis. The number of dead cells was normalized by the initial cell number at time 0, giving the percentage of cell death as plotted in the new Figure 5. The data were averaged from two independent imaging experiments and the total number of cells analyzed ranged from 66 to 256, varied between conditions and cell lines. In general, the simulation and experimental results showed similar response profiles across the different cell lines and treatment conditions. However, significant discrepancy was observed in CAL51 and SJSA1 cells under etoposide treatment alone, where the simulation analysis over-predicted the cell death response for SJSA1 and under-predicted the cell death response for CAL51, as compared to experimental results from time-lapse imaging. This discrepancy could potentially be due to: 1) incomplete information from the genomic data that limits the accuracy of network mapping and subsequent network simulation analysis; and/or 2) the use of a simplified p53 network for our network analysis, where genomic alterations of additional p53 network components that are not included in our network model may play an important role in regulating drug response. We added the above discussion in page 20 of the revised manuscript.

[COMMENT #3].

“There are also some restrictions inherent to the approach of the authors. As they focus on an abstracted p53 network in p53 wt cells, alterations that are not part of the initial network and could represent truly “non-intuitive” targets are neglected. Also, p53 mutations that often

acquire new functions, are also not included in the analysis. The use of a boolean network together with discretised genomic alterations limit the predictive power. For example, the authors can't distinguish between weakly and strongly activating mutations or consider varying efficiencies of inhibitors. While these limitations are inherent to the approach, the authors should at least discuss them openly.”

[RESPONSE] We agree the simplified p53 network model that we used as an example of the network dynamics-based approach is limited in the aspects that the reviewers mentioned. However, our computational approach is still widely applicable to model complex regulatory networks, where continuous modeling is not appropriate or possible, due to insufficient quantitative information on kinetic parameters^{1,2}. And we do think our approach provides a foundational framework that can be further developed to address the limitations pointed out by the reviewer. For instance, we can easily expand the simplified p53 network to a larger, more comprehensive functional network that incorporates additional components crucial to oncogenesis, metastasis and/or tumor response. We could also expand the Boolean network model to multi-valued logical model or use fuzzy logic, instead of Boolean logic, to describe varying degrees of activation/inhibition and drug efficacy^{2,3}. As for cancer cells with mutant p53, the drug effect is likely mediated by alternative signaling pathway so the network dynamics-based analysis should probably be developed and performed other signaling pathway, beyond the p53 network. We added the above discussion regarding the limitation of our approach in page 20 of the revised manuscript.

[COMMENT #4].

Minor points:

“* Figure legends are extremely brief and lack detail. There are many small issues with the figures:

* Figure 1: no explanation of e.g. node color after mapping of alterations is given, panel b is not described at all.”

[RESPONSE] We rewrote the figure legend of Figure 1 and added the missing information. In Figure 1a, the node color represents the status of the node activity. A black (white) node means that the node is constantly activated (inactivated) and a gray node means that the status is dependent on the activity of a given input. Figure 1b illustrates the general framework of our method to predict drug response using attractor landscape analysis of network dynamics. In the revised figure legend, we describe the p53 network model, selection of drug target and definition of response phenotype based on attractor states.

“* Figure 2: abbreviations such as HOMDEL, AMP etc should be spelled out at least once; heat maps in panel a should contain labels and there are typos in y-axis label for the bar graph next to pie charts ("Altered Smamples").”

[RESPONSE] We corrected the gene names in Figure 2 and also added a full description of “HOMDEL” and “AMP” in the revised manuscript and Figure legend.

“* Figure 3: there are no numbers on scale for heat map.”

[RESPONSE] We added numbers on scale for heat map in the revised figure.

“* The naming switches between genes and nodes, e.g. CDKN2A - p14ARF, PPM1D - Wip1 in Figure 2. Here a consistent labelling would be easier for the reader.”

[RESPONSE] In the revised manuscript, we consistently used protein names to describe the nodes. For example, CDKN2A and PPM1D have been changed to p14ARF and Wip1 throughout the manuscript.

References

1. Wang RS, Saadatpour A, Albert R. Boolean modeling in systems biology: an overview of methodology and applications. *Phys Biol* **9**, 055001 (2012).
2. Abou-Jaoude W, *et al.* Logical Modeling and Dynamical Analysis of Cellular Networks. *Front Genet* **7**, 94 (2016).
3. Aldridge BB, Saez-Rodriguez J, Muhlich JL, Sorger PK, Lauffenburger DA. Fuzzy logic analysis of kinase pathway crosstalk in TNF/EGF/insulin-induced signaling. *PLoS Comput Biol* **5**, e1000340 (2009).

Reviewers' comments:

Reviewer #1 (Remarks to the Author):

The authors have addressed this reviewers comments.

Reviewer #2 (Remarks to the Author):

Comment #1:

In general, the authors have improved the clarity and accessibility of the manuscript.

Comment #2:

I appreciate the effort to expand the experimental validation from 3 to 8 cell lines and to provide a more solid quantification of cell death using time-lapse microscopy. However, I am not entirely sure how to interpret these results. For 2 of 5 new cell lines, predictions for etoposide alone are off as admitted by the authors. I would argue that at least for Cal51, not only the response to etoposide alone is off, but the predictions of 7 out of 13 perturbations significantly deviate from experimental results. When the two cell lines of network subgroup 3 are compared (A2780 and Cal51), one notices that predictions are identical, but experimental results vary qualitatively (e.g. for B_N, N_W, E_N). It almost seems that the overall predictions for these two lines would be as good (or as bad) if the model of another cell line (e.g. MCF7) was used. As mentioned in my initial comment, conclusions from the experimental validation would be stronger if the authors find a way to quantify the deviation of model predictions and experimental data and, if possible, provide some insight into how much better a cell line specific prediction is compared to "random" predictions, e.g. generated by shuffling predictions for the corresponding perturbation for each cell line. This would also emphasise "surprising" cell line specific prediction. I also need to mention that the authors still limit experimental validation to only one of the predicted outcomes (cell death), and ignore the other two (cell cycle arrest and proliferation).

Comment #3:

The authors now provide a more satisfying discussion of model limitations and potential improvements.

Response to the Reviewer's Comments and Summary of Changes

Manuscript ID: NCOMMS-16-30137A

Title: Network dynamics-based stratification of cancer panel for systemic prediction of anticancer drug response

Authors: Minsoo Choi, Jue Shi, Yanting Zhu, Ruizhen Yang and Kwang-Hyun Cho

Response to the specific comments of Reviewer 2:

[COMMENT #1].

“I appreciate the effort to expand the experimental validation from 3 to 8 cell lines and to provide a more solid quantification of cell death using time-lapse microscopy. However, I am not entirely sure how to interpret these results. For 2 of 5 new cell lines, predictions for etoposide alone are off as admitted by the authors. I would argue that at least for Cal51, not only the response to etoposide alone is off, but the predictions of 7 out of 13 perturbations significantly deviate from experimental results. When the two cell lines of network subgroup 3 are compared (A2780 and Cal51), one notices that predictions are identical, but experimental results vary qualitatively (e.g. for B_N, N_W, E_N). It almost seems that the overall predictions for these two lines would be as good (or as bad) if the model of another cell line (e.g. MCF7) was used.

As mentioned in my initial comment, conclusions from the experimental validation would be stronger if the authors find a way to quantify the deviation of model predictions and experimental data and, if possible, provide some insight into how much better a cell line specific prediction is compared to “random” predictions, e.g. generated by shuffling predictions for the corresponding perturbation for each cell line. This would also emphasize “surprising” cell line specific prediction. I also need to mention that the authors still limit experimental validation to only one of the predicted outcomes (cell death), and ignore the other two (cell cycle arrest and proliferation).”

[RESPONSE]: We again thank the reviewer for the very helpful comments. Regarding the reviewer's major concern of a lack of quantitative analysis of the deviation of modeling predictions and experimental data, we have performed new statistical analysis in the revised manuscript and added them to the new Figure 5. Specifically, we calculated the Pearson correlation coefficients and the root mean square errors (RMSEs) between the experimentally observed and predicted drug responses of each cell line under the 13 different drug treatment

conditions (refer to the new Figure 5 in the revised manuscript). The overall Pearson correlation across all cell lines combined is high (correlation coefficient: 0.75, $p < 0.001$), indicating that the drug responses are in general well predicted by our network dynamics analysis. Moreover, RMSE between the experimentally observed and predicted drug responses in a given cell line was calculated as follows:

$$RMSE(C, O, P) = \sqrt{\frac{1}{n} \sum_{i=1}^n (O_i - P_i)^2}$$

where n is the number of total drug treatment conditions (i.e., 13), O_i is the observed cell death ratio of a given cell line C for a treatment i , and P_i is the predicted cell death ratio of a given cell line C for a treatment i . In the revised manuscript, we plotted the values of $1/RMSE$ with respect to the Pearson correlation coefficients for each cell line in the new Figure 5d, which illustrates that our method performs quite well in both statistical measures, in particular for cell lines at the upper right-hand corner of the graph.

However, we do note there is significant discrepancy between some modeling and experimental results, as rightly pointed out by the reviewer. For example, a relatively high RMSE of CAL51 was observed as compared to the other 7 cancer cell lines, even though the Pearson correlation of CAL51 was high. For CAL51, the experimentally observed cell death responses were larger than the predicted responses under all drug treatment conditions, including DNA damaging drug alone (E). We thus suspect the discrepancy may arise from deficiency of the CAL51 network that we curated due to: 1) incomplete information from the genomic data that limits the accuracy of network mapping and subsequent network simulation analysis; and/or 2) the use of a simplified p53 network in our study, where genomic alterations of additional p53 network components that are not included in our network model may play an important role in regulating drug responses. For instance, our CAL51 network model does not include some critical genomic alterations observed in CAL51, such as mutations of MAP2K4 and CHEK1. MAP2K4 is known to activate JUN kinase and p38 signaling pathways and is involved in p53-mediated response of cell cycle arrest and cell death¹. CHEK1 is also known to coordinate cell cycle checkpoint response and DNA damage response. Mutation of CHEK1 could lead to the loss of CHK1, resulting in increased sensitivity to DNA damaging agents, such as etoposide². We added discussion of this limitation of our model in detail in page 18-19 of the revised manuscript.

Regarding the point raised by the reviewer that our simulation results showed very similar perturbation responses for A2780 and CAL51, we think this is probably due to the fact that the only difference between A2780 and CAL51 under the simplified p53 network that we employed is MDMX, which is constantly activated in A2780, but not in CAL51. And both networks share two key alterations, including constantly inactivated PTEN and constantly activated AKT (i.e., PTEN(I) and AKT(A)) (refer to Fig. R1 and R2 below). In our p53 network model, MDMX has two activating inputs, AKT and Wip1, and three inhibitory inputs, ATM, p14ARF and MDM2. The state of MDMX is regulated by interaction of these five inputs, therefore network topology alone cannot predict MDMX status under the different drug treatment conditions. To specifically examine the effect of constantly activated MDMX (MDMX(A)), we investigated the attractor states of A2780 and CAL51. We found that A2780 and CAL51 have the same attractor states, irrespective of the MDMX status, although each basin of attraction is slightly different. This modeling result indicates that in the presence of constantly activated PTEN and AKT, MDMX does not significantly affect the extent of cell death response, which is determined by the overall network dynamics (Fig. R1). In addition, MDMX is always “ON” in all attractor states of CAL51 under the distinct perturbations, even though MDMX is not constantly activated in the control CAL51 network without inhibitory perturbation. This indicates that AKT(A) is a dominant input that keeps MDMX “ON” in the CAL51-specific network model that we curated, despite that MDMX has multiple upstream regulators (Fig. R2).

CAL51

Inhibitory treatment	ATM	R53	Mdm2	Mdmx	Wip1	Cyc	PTEN	R21	AKT	Cyc	PRB	E2F1	ARF	Bcl2	Caspase	Basin size (Ratio)
B																0.05542
B																0.02472
B																0.00015
N																0.05994
W																0.04446
W																0.00006
B_W																0.08636
B_W																0.00015
B_N																0.14822
N_W																0.57788
N_W																0.03592
N_W																0.01257
N_W																0.00989
E_B																0.14151
E_B																0.10196
E_B																0.02954
E_B																0.00220
E_B																0.00128
E_B																0.00110
E_B																0.00067
E_B																0.00058
E_N																0.45346
E_N																0.00491
E_N																0.00336
E_N																0.00061
E_N																0.00052
E_N																0.00040
E_N																0.00037
E_N																0.00031
E_N																0.00021
E_W																0.01981
E_W																0.00952
E_B_W																0.04797
E_B_W																0.03687
E_B_N																0.49249
E_B_N																0.20395
E_B_N																0.00818
E_B_N																0.00079
E_B_N																0.00067
E_B_N																0.00055
E_B_N																0.00049
E_B_N																0.00037
E_B_N																0.00027
E_N_W																0.39209
E_N_W																0.16431
E_N_W																0.08154
E_N_W																0.07437
E																0.06406
E																0.01096
E																0.00131
E																0.00125
E																0.00095
E																0.00082
E																0.00052
E																0.00046

A2780

Inhibitory treatment	ATM	R53	Mdm2	Mdmx	Wip1	Cyc	PTEN	R21	AKT	Cyc	PRB	E2F1	ARF	Bcl2	Caspase	Basin size (Ratio)
B																0.04120
B																0.02097
B																0.00037
N																0.04343
W																0.03427
W																0.00027
B_W																0.06424
B_W																0.00037
B_N																0.09601
N_W																0.58862
N_W																0.03180
N_W																0.00867
N_W																0.00812
E_B																0.12241
E_B																0.10831
E_B																0.02277
E_B																0.00574
E_B																0.00128
E_B																0.00110
E_B																0.00067
E_B																0.00058
E_N																0.48074
E_N																0.00650
E_N																0.00519
E_N																0.00061
E_N																0.00058
E_N																0.00052
E_N																0.00037
E_N																0.00031
E_N																0.00027
E_W																0.02582
E_W																0.01089
E_B_W																0.05225
E_B_W																0.04559
E_B_N																0.52118
E_B_N																0.14798
E_B_N																0.01044
E_B_N																0.00079
E_B_N																0.00079
E_B_N																0.00067
E_B_N																0.00049
E_B_N																0.00037
E_B_N																0.00034
E_N_W																0.43628
E_N_W																0.15552
E_N_W																0.05762
E_N_W																0.04834
E																0.06924
E																0.01071
E																0.00403
E																0.00262
E																0.00095
E																0.00082
E																0.00052
E																0.00046

Figure R1. Cell death attractor states and their basin size (ratio) under different network perturbations in CAL51 (left) and A2780 (right) specific networks. Each attractor state is represented by a set of 16 boxes in each row. Each box represents the state of each network node in the attractor state. A navy box or white box indicates “ON” or “OFF”, respectively. Two long red boxes highlight MDMX and Caspase.

Figure R2. PTEN-AKT-MDMX interactions in the p53 network models. MDMX has two activating upstream regulators, AKT, Wip1 and three inhibitory upstream regulators, ATM, p14ARF and MDM2 in our model. The interactions are denoted by black dotted lines. The black solid line denotes a constantly activated link and the gray dotted lines denote insignificant links, as they are either from constantly inactivated nodes or their targets are constantly activated or inactivated nodes. In the absence of network alteration, the state of MDMX is determined by combination of all five inputs. However, in the presence of node alterations, where PTEN is constantly inactivated and AKT is constantly activated (as in both the CAL51 and A2780 specific networks), AKT(A) becomes a dominant input that keeps MDMX “ON”, despite that MDMX has multiple upstream regulators. Therefore, the difference of MDMX state in the CAL51 and A2780 networks does not significantly affect the cell death response of these two cell lines, which is determined by the overall network dynamics in our modeling analysis.

Moreover, to further examine the predictive power of our network dynamics analysis, we compared our cell line specific predictions with random predictions acquired by shuffling alterations of each cell line such that the number of alterations is preserved, while their locations are randomized (refer to the new Fig. 5e and 5f in the revised manuscript). Briefly, 480 and 4480 random networks, which are all possible networks that have two node or three node alterations, were generated by shuffling the alterations of A375, 769P, CAL51 which have two node alterations and A549, MCF7, U2OS, SJSA1 and A2780 which have three node alterations, respectively. We defined the RMSE between the experimentally observed and randomly predicted drug responses in a given cell line as follows:

$$RMSE(C, O, R) = \frac{1}{m} \sum_{j=1}^m \sqrt{\frac{1}{n} \sum_{i=1}^n (O_i - R_{i,j})^2}$$

where n is the number of total drug treatment conditions (i.e., 13), m is the number of random networks, O_i is the observed cell death ratio of a given cell line C for a treatment i , $R_{i,j}$ is the predicted cell death ratio of random network j for a treatment i . For all random predictions, we observed relatively weak correlations between the experimentally observed and randomly predicted responses, compared to that between the experimental data and predicted responses by our approach ($p < 0.001$, Wilcoxon rank sum test). In addition, each RMSE of our cell line specific predictions is significantly smaller than that of the random prediction ($p < 0.001$, Wilcoxon rank sum test). Overall, these results demonstrate that our network dynamics analysis performs substantially better than random prediction. We added this discussion in page 19 of the revised manuscript.

In this study, we chose to focus on performing detailed analysis of the cell death response because clinical response of tumor regression or delay in tumor growth often correlates with the cell death response and variation in cell death response to cancer chemotherapy is likely the most crucial factor that distinguishes sensitive and resistant tumors.

Reference

1. Keshet Y, Seger R. The MAP kinase signaling cascades: a system of hundreds of components regulates a diverse array of physiological functions. *Methods Mol Biol* **661**, 3-38 (2010).
2. Cho SH, Toouli CD, Fujii GH, Crain C, Parry D. Chk1 is essential for tumor cell viability following activation of the replication checkpoint. *Cell Cycle* **4**, 131-139 (2005).

REVIEWERS' COMMENTS:

Reviewer #2 (Remarks to the Author):

The authors have satisfactorily addressed the remaining concern.